# EXPLORATION BY RANDOM NETWORK DISTILLATION

## ABSTRACT

We introduce an exploration bonus for deep reinforcement learning methods that is easy to implement and adds minimal overhead to the computation performed. The bonus is the error of a neural network predicting features of the observations given by a fixed randomly initialized neural network. We also introduce a method to flexibly combine intrinsic and extrinsic rewards. We find that the random network distillation (RND) bonus combined with this increased flexibility enables significant progress on several hard exploration Atari games. In particular we establish state of the art performance on Montezuma's Revenge, a game famously difficult for deep reinforcement learning methods. To the best of our knowledge, this is the first method that achieves better than average human performance on this game without using demonstrations or having access to the underlying state of the game, and occasionally completes the first level. This suggests that relatively simple methods that scale well can be sufficient to tackle challenging exploration problems.

## 1 INTRODUCTION

Reinforcement learning (RL) methods work by maximizing the expected return of a policy. This works well when the environment has dense rewards that are easy to find by taking random sequences of actions, but tends to fail when the rewards are sparse and hard to find. In reality it is often impractical to engineer dense reward functions for every task one wants an RL agent to solve. In these situations methods that explore the environment in a directed way are necessary.

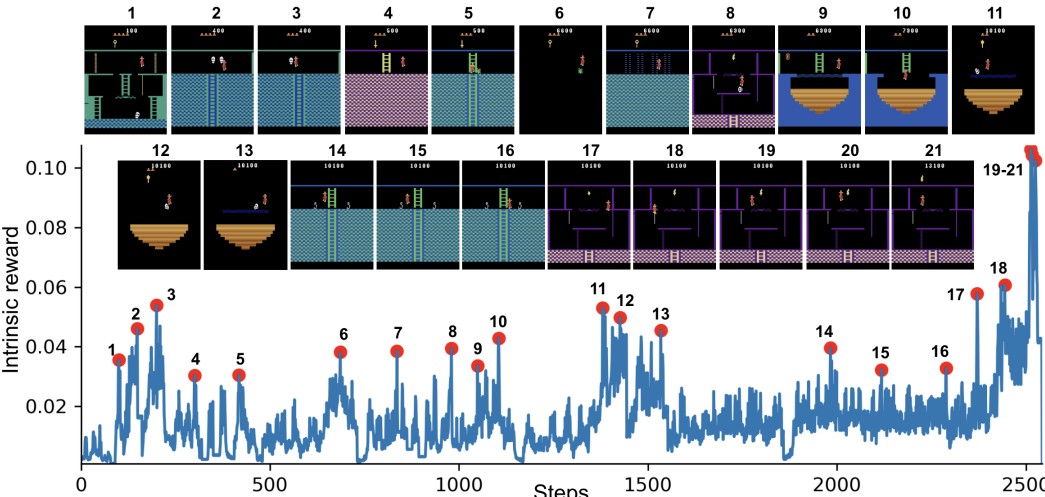

Figure 1: RND exploration bonus over the course of the first episode where the agent picks up the torch (19-21). To do so the agent passes 17 rooms and collects gems, keys, a sword, an amulet, and opens two doors. Many of the spikes in the exploration bonus correspond to meaningful events: losing a life (2,8,10,21), narrowly escaping an enemy (3,5,6,11,12,13,14,15), passing a difficult obstacle (7,9,18), or picking up an object (20,21). The large spike at the end corresponds to a novel experience of interacting with the torch, while the smaller spikes correspond to relatively rare events that the agent has nevertheless experienced multiple times. See goo.gl/DGPC8E for videos.

Recent developments in RL seem to suggest that solving the most challenging tasks (Silver et al., 2016; Zoph & Le, 2016; Horgan et al., 2018; Espeholt et al., 2018; OpenAI, 2018; OpenAI et al., 2018) requires processing large numbers of samples obtained from running many copies of the environment in parallel. In light of this it is desirable to have exploration methods that scale well with large amounts of experience. However many of the recently introduced exploration methods based on counts, pseudo-counts, information gain or prediction gain are difficult to scale up to large numbers of parallel environments.

This paper introduces an exploration bonus that is particularly simple to implement, works well with high-dimensional observations, can be used with any policy optimization algorithm, and is efficient to compute as it requires only a single forward pass of a neural network on a batch of experience. Our exploration bonus is based on the observation that neural networks tend to have significantly lower prediction errors on examples similar to those on which they have been trained. This motivates the use of prediction errors of networks trained on the agent's past experience to quantify the novelty of new experience.

As pointed out by many authors, agents that maximize such prediction errors tend to get attracted to transitions where the answer to the prediction problem is a stochastic function of the inputs. For example if the prediction problem is that of predicting the next observation given the current observation and agent's action (forward dynamics), an agent trying to maximize this prediction error will tend to seek out stochastic transitions, like those involving randomly changing static noise on a TV, or outcomes of random events such as coin tosses. This observation motivated the use of methods that quantify the relative improvement of the prediction, rather than its absolute error. Unfortunately, as previously mentioned, such methods are hard to implement efficiently.

We propose an alternative solution to this undesirable stochasticity by defining an exploration bonus using a prediction problem where the answer is a deterministic function of its inputs. Namely we predict the output of a fixed randomly initialized neural network on the current observation.

Atari games have been a standard benchmark for deep reinforcement learning algorithms since the pioneering work by Mnih et al. (2013). Bellemare et al. (2016) identified among these games the hard exploration games with sparse rewards: Freeway, Gravitar, Montezuma's Revenge, Pitfall!, Private Eye, Solaris, and Venture. RL algorithms tend to struggle on these games, often not finding even a single positive reward.

In particular, Montezuma's Revenge is considered to be a difficult problem for RL agents, requiring a combination of mastery of multiple in-game skills to avoid deadly obstacles, and finding rewards that are hundreds of steps apart from each other even under optimal play. Significant progress has been achieved by methods with access to either expert demonstrations (Pohlen et al., 2018; Aytar et al., 2018; Garmulewicz et al., 2018), special access to the underlying emulator state (Tang et al., 2017; Stanton & Clune, 2018), or both (Salimans & Chen, 2018). However without such aids, progress on the exploration problem in Montezuma's Revenge has been slow, with the best methods finding about half the rooms (Bellemare et al., 2016). For these reasons we provide extensive ablations of our method on this environment.

We find that even when disregarding the extrinsic reward altogether, an agent maximizing the RND exploration bonus consistently finds more than half of the rooms in Montezuma's Revenge. To combine the exploration bonus with the extrinsic rewards we introduce a modification of Proximal Policy Optimization (PPO, Schulman et al. (2017)) that uses two value heads for the two reward streams. This allows the use of different discount rates for the different rewards, and combining episodic and non-episodic returns. With this additional flexibility, our best agent often finds 22 out of the 24 rooms on the first level in Montezuma's Revenge, and occasionally (though not frequently) passes the first level. The same method gets state of the art performance on Venture and Gravitar.

## 2 METHOD

### 2.1 EXPLORATION BONUSES

Exploration bonuses are a class of methods that encourage an agent to explore even when the environment's reward $e_t$ is sparse. They do so by replacing $e_t$ with a new reward $r_t = e_t + i_t$, where $i_t$ is the exploration bonus associated with the transition at time $t$.

To encourage the agent to visit novel states, it is desirable for $i_t$ to be higher in novel states than in frequently visited ones. Count-based exploration methods provide an example of such bonuses.

In a tabular setting with a finite number of states one can define $i_t$ to be a decreasing function of the visitation count $n_t(s)$ of the state $s$. In particular $i_t = 1/n_t(s)$ and $i_t = 1/\sqrt{n_t(s)}$ have been used in prior work (Bellemare et al., 2016; Ostrovski et al., 2018). In non-tabular cases it is not straightforward to produce counts, as most states will be visited at most once. One possible generalization of counts to non-tabular settings is pseudo-counts (Bellemare et al., 2016) which uses changes in state density estimates as an exploration bonus. In this way the counts derived from the density model can be positive even for states that have not been visited in the past, provided they are similar to previously visited states.

An alternative is to define $i_t$ as the prediction error for a problem related to the agent's transitions. Generic examples of such problems include forward dynamics and inverse dynamics (Schmidhuber, 1991b; Stadie et al., 2015; Achiam & Sastry, 2017; Pathak et al., 2017; Burda et al., 2018; Haber et al., 2018). Non-generic prediction problems can also be used if specialized information about the environment is available, like predicting physical properties of objects the agent interacts with (Denil et al., 2016). Such prediction errors tend to decrease as the agent collects more experience similar to the current one. For this reason even trivial prediction problems like predicting a constant zero function can work as exploration bonuses (Fox et al., 2018).

## 2.2 RANDOM NETWORK DISTILLATION

This paper introduces a different approach where the prediction problem is randomly generated. This involves two neural networks: a fixed and randomly initialized *target* network which sets the prediction problem, and a *predictor* network trained on data collected by the agent. The target network takes an observation to an embedding $f : \mathcal{O} \to \mathbb{R}^k$ and the predictor neural network $\hat{f} : \mathcal{O} \to \mathbb{R}^k$ is trained by gradient descent to minimize the expected MSE $\|\hat{f}(\mathrm{x}; \theta) - f(\mathrm{x})\|^2$ with respect to its parameters $\theta_{\hat{f}}$. This process distills a randomly initialized neural network into a trained one. The prediction error $i_t = \|\hat{f}(\mathrm{x}) - f(\mathrm{x})\|^2$ is expected to be higher for novel states dissimilar to the ones the predictor has been trained on. This allows to use $i_t$ as an exploration bonus.

To build intuition we consider a toy model of this process on MNIST. We train a predictor neural network to mimic a randomly initialized target network on training data consisting of a mixture of images with the label 0 and of a target class, varying the proportion of the classes, but not the total number of training examples. We then test the predictor network on the unseen test examples of the target class and report the MSE. In this model the zeros are playing the role of states that have been seen many times before, and the target class is playing the role of states that have been visited infrequently. The results are shown in Figure 2. The figure shows that test error decreases as a function of the number of training examples in the target class, suggesting that this method can be used to detect novelty. Figure 1 shows that the intrinsic reward is high in novel states in an episode of Montezuma's Revenge.

One objection to this method is that a sufficiently powerful optimization algorithm might find a predictor that mimics the target random network perfectly on any input (for example the target network itself would be such a predictor). However the above experiment on MNIST shows that standard gradient-based methods don't overgeneralize in this undesirable way.

### 2.2.1 SOURCES OF PREDICTION ERRORS

In general, prediction errors can be attributed to a number of factors:

1. *Amount of training data*. Prediction error is high where few similar examples were seen by the predictor (epistemic uncertainty).

2. *Stochasticity*. Prediction error is high because the target function is stochastic (aleatoric uncertainty). Stochastic transitions are a source of such error for forward dynamics prediction.

3. *Model misspecification*. Prediction error is high because necessary information is missing, or the model class is too limited to fit the complexity of the target function.

4. *Learning dynamics*. Prediction error is high because the optimization process fails to find a predictor in the model class that best approximates the target function.

Factor 1 is what allows one to use prediction error as an exploration bonus. In practice the prediction error is caused by a combination of all of these factors, not all of them desirable.

For instance if the prediction problem is forward dynamics, then factor 2 results in the 'noisy-TV' problem. This is the thought experiment where an agent that is rewarded for errors in the prediction of its forward dynamics model gets attracted to stochastic transitions in the environment. A TV randomly switching between channels would be such an attractor, as would a coin flip.

To avoid the undesirable factors 2 and 3, methods such as those by Schmidhuber (1991a); Oudeyer et al. (2007); Lopes et al. (2012); Achiam & Sastry (2017) instead use a measurement of how much the prediction model improves upon seeing a new datapoint. However these approaches tend to be computationally expensive and hence difficult to scale.

RND obviates factors 2 and 3 since the target network can be chosen to be deterministic and inside the model-class of the predictor network.

### 2.2.2 RELATION TO UNCERTAINTY QUANTIFICATION

In this section we highlight a link between the RND prediction error and an uncertainty quantification method introduced by Osband et al. (2018). Namely, consider a regression problem with data distribution $D = \{x_i, y_i\}_i$. In the Bayesian setting we would consider a prior $p(\theta^*)$ over the parameters of a mapping $f_{\theta^*}$ and calculate the posterior after updating on the evidence.

Let $\mathcal{F}$ be the distribution over functions $g_\theta = f_\theta + f_{\theta^*}$, where $\theta^*$ is drawn from $p(\theta^*)$ and $\theta$ is given by minimizing the expected prediction error

$$\theta = \arg\min_\theta \mathbb{E}_{(x_i, y_i) \sim D} \|f_\theta(x_i) + f_{\theta^*}(x_i) - y_i\|^2 + \mathcal{R}(\theta), \tag{1}$$

where $\mathcal{R}(\theta)$ is a regularization term coming from the prior (see Lemma 3, Osband et al. (2018)). Osband et al. (2018) argue that the ensemble $\mathcal{F}$ is an approximation of the posterior. In the case of Bayesian linear regression this statement can be made precise. However even in the case where the functions are not linear, (Osband et al., 2018) experimentally validate that the same procedure can be used as a part of a heuristic for quantifying uncertainty.

If we specialize the regression targets $y_i$ to be zero, then the optimization problem $\arg\min_\theta \mathbb{E}_{(x_i, y_i) \sim D} \|f_\theta(x_i) + f_{\theta^*}(x_i)\|^2$ is equivalent to distilling a randomly drawn function from the prior. (Here we omit the regularization term from the objective and assume that the prior is symmetric around the origin in the parameter space). Seen from this perspective, each coordinate of the output of the predictor and target networks would correspond to a member of an ensemble (with parameter sharing amongst the ensemble), and the MSE would be an estimate of the predictive variance of the ensemble (assuming the ensemble is unbiased). In other words the distillation error could be seen as a quantification of uncertainty in predicting the constant zero function. We believe that a similar mechanism might underlie the performance of RND and (Osband et al., 2018).

### 2.3 COMBINING INTRINSIC AND EXTRINSIC RETURNS

In preliminary experiments that used only intrinsic rewards, treating the problem as non-episodic resulted in better exploration. In that setting the return is not truncated at "game over". We argue that this is a natural way to do exploration in simulated environments, since the agent's intrinsic return should be related to all the novel states that it could find in the future, regardless of whether they all occur in one episode or are spread over several. It is also argued in (Burda et al., 2018) that using episodic intrinsic rewards can leak information about the task to the agent.

We also argue that this is closer to how humans explore games. For example let's say Alice is playing a videogame and is attempting a tricky maneuver to reach a suspected secret room. Because the maneuver is tricky the chance of a game over is high, but the payoff to Alice's curiosity will be high if she succeeds. If Alice is modelled as an episodic reinforcement learning agent, then her future return will be exactly zero if she gets a game over, which might make her overly risk averse. The real cost of a game over to Alice is the opportunity cost incurred by having to play through the game from the beginning (which is presumably less interesting to Alice having played the game for some time).

However using non-episodic returns for extrinsic rewards could be exploited by a strategy that finds a reward close to the beginning of the game, deliberately restarts the game by getting a game over, and repeats this in an endless cycle.

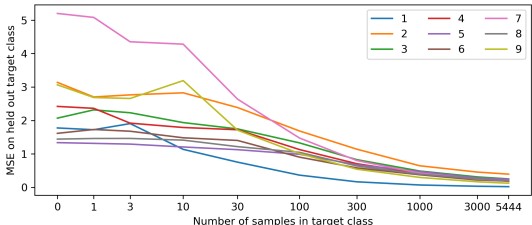 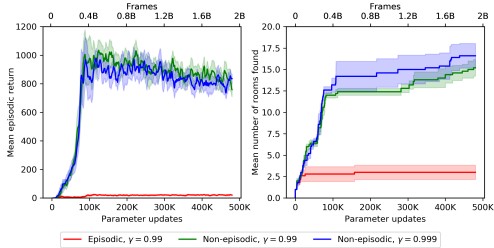

Figure 2: Novelty detection on MNIST: a predictor network mimics a randomly initialized target network. The training data consists of varying proportions of images from class "0" and a target class. Each curve shows the test MSE on held out target class examples plotted against the number of training examples of the target class (log scale). Curves are an average over 10 random seeds.

Figure 3: Mean episodic return and number of rooms found by pure exploration agents on Montezuma's Revenge trained without access to the extrinsic reward. The agents explores more in the non-episodic setting (see also Section 2.3). Curves are an average over 5 random seeds.

It is not obvious how to estimate the combined value of the non-episodic stream of intrinsic rewards $i_t$ and the episodic stream of extrinsic rewards $e_t$. Our solution is to observe that the return is linear in the rewards and so can be decomposed as a sum $R = R_E + R_I$ of the extrinsic and intrinsic returns respectively. Hence we can fit two value heads $V_E$ and $V_I$ separately using their respective returns, and combine them to give the value function $V = V_E + V_I$. This same idea can also be used to combine reward streams with different discount factors.

Note that even where one is not trying to combine episodic and non-episodic reward streams, or reward streams with different discount factors, there may still be a benefit to having separate value functions since there is an additional supervisory signal to the value function. This may be especially important for exploration bonuses since the extrinsic reward function is stationary whereas the intrinsic reward function is non-stationary.

## 3 EXPERIMENTS

We begin with an intrinsic reward only experiment on Montezuma's Revenge in Section 3.1 to isolate the inductive bias of the RND bonus, follow by extensive ablations of RND on Montezuma's Revenge in Sections 3.2-3.5 to understand the factors that contribute to RND's performance, and conclude with a comparison to baseline methods on 6 hard exploration Atari games in Section 3.6. For details of hyperparameters and architectures we refer the reader to Appendices A.3 and A.4. Most experiments are run for 30K rollouts of length 128 per environment with 128 parallel environments, for a total of 1.97 billion frames of experience. Each curve is an average over a number of random seeds detailed in the caption, and the shaded region is a standard error. Both the mean and the standard error curves were smoothed by averaging over a sliding window of 1.3% of the datapoints to make the figures more legible. We use the PPO (Schulman et al., 2017) as our policy optimization algorithm for all experiments.

### 3.1 PURE EXPLORATION

In this section we explore the performance of RND in the absence of any extrinsic reward. In Section 2.3 we argued that exploration with RND might be more natural in the non-episodic setting. By comparing the performance of the pure exploration agent in episodic and non-episodic settings we can see if this observation translates to improved exploration performance.

We report two measures of exploration performance in Figure 3: mean episodic return, and the number of rooms the agent finds over the training run. Since the pure exploration agent is not aware of the extrinsic rewards or number of rooms, it is not directly optimizing for any of these measures. However obtaining some rewards in Montezuma's Revenge (like getting the key to open a door) is required for accessing more interesting states in new rooms, and hence we observe the extrinsic reward increasing over time up to some point. The best return is achieved when the agent interacts with some of the objects, but the agent has no incentive to keep doing the same once such interactions become repetitive, hence returns are not consistently high.

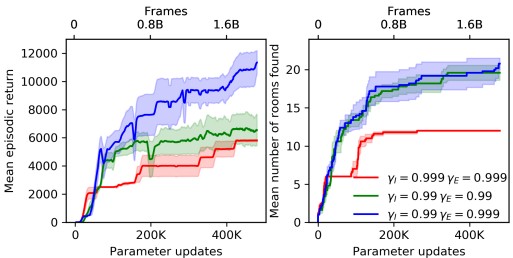 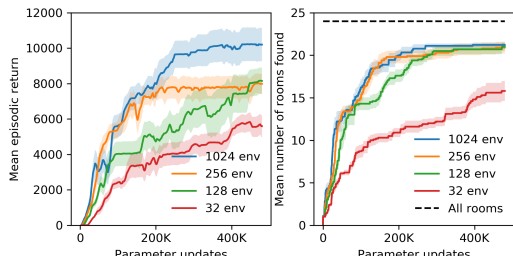

Figure 4: Performance of different discount factors for intrinsic and extrinsic reward streams. A higher discount factor for the extrinsic rewards leads to better performance, while for intrinsic rewards it hurts exploration. Curves are an average over 5 random seeds.

Figure 5: Mean episodic return and number of discovered rooms improve as the number of parallel environments used for collecting the experience increases. The runs have processed 0.5,2,4, and 16B frames. Curves are an average over 10 random seeds.

We clearly see in Figure 3 that on both measures of exploration the non-episodic agent performs best, consistent with the discussion in Section 2.3. The non-episodic setting with $\gamma_I = 0.999$ explores more rooms than $\gamma_I = 0.99$, with one of the runs exploring 21 rooms. The best return achieved by 4 out 5 runs of this setting was 6,700.

## 3.2 COMBINING EPISODIC AND NON-EPISODIC RETURNS

In Section 3.1 we saw that the non-episodic setting resulted in more exploration than the episodic setting when exploring without any extrinsic rewards. Next we consider whether this holds in the case where we combine intrinsic and extrinsic rewards. As discussed in Section 2.3 in order to combine episodic and non-episodic reward streams we require two value heads. This also raises the question of whether it is better to have two value heads even when both reward streams are episodic. In Figure 6 we compare episodic intrinsic rewards to non-episodic intrinsic rewards combined with episodic extrinsic rewards, and additionally two value heads versus one for the episodic case. The discount factors are $\gamma_I = \gamma_E = 0.99$.

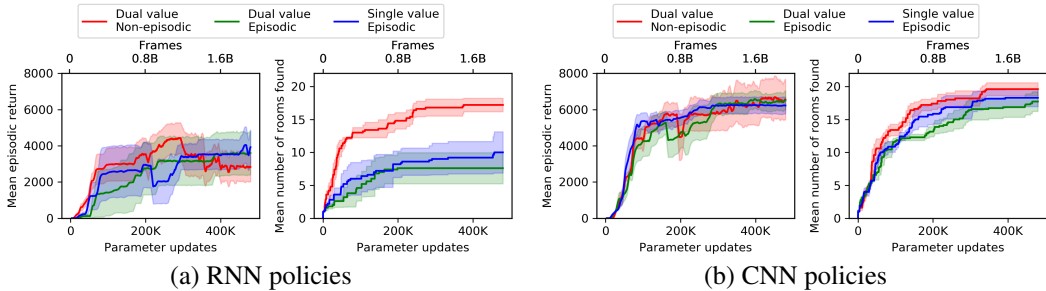

(a) RNN policies          (b) CNN policies

Figure 6: Different ways of combining intrinsic and extrinsic rewards. Combining non-episodic stream of intrinsic rewards with the episodic stream of extrinsic rewards outperforms combining episodic versions of both steams in terms of number of explored rooms, but performs similarly in terms of mean return. Single value estimate of the combined stream of episodic returns performs a little better than the dual value estimate. The differences are more pronounced with RNN policies. CNN runs are more stable than the RNN counterparts. Curves are an average over 5 random seeds.

In Figure 6 we see that using a non-episodic intrinsic reward stream increases the number of rooms explored for both CNN and RNN policies, consistent with the experiments in Section 3.1, but that the difference is less dramatic, likely because the extrinsic reward is able to preserve useful behaviors. We also see that the difference is less pronounced for the CNN experiments, and that the RNN results tend to be less stable and perform worse overall.

Contrary to our expectations (Section 2.3) using two value heads did not show any benefit over a single head in the episodic setting. Nevertheless having two value heads is necessary for combining reward streams with different characteristics (for example having different discount factors or combining episodic rewards with non-episodic reward), and so all further experiments use two value heads.

### 3.3 DISCOUNT FACTORS

Previous experiments (Salimans & Chen, 2018; Pohlen et al., 2018; Garmulewicz et al., 2018) solving Montezuma's Revenge using expert demonstrations used a high discount factor to achieve the best performance, enabling the agent to anticipate rewards far into the future. We compare the performance of the RND agent with $\gamma_E \in \{0.99, 0.999\}$ and $\gamma_I = 0.99$. We also investigate the effect of increasing $\gamma_I$ to 0.999. The results are shown in Figure 4.

In Figure 4 we see that increasing $\gamma_E$ to 0.999 while holding $\gamma_I$ at 0.99 greatly improves performance. This setting had a mean return of 11.5K at the end of training, setting a new state of the art. We also see that further increasing $\gamma_I$ to 0.999 hurts performance. This is at odds with the results in Figure 3 where increasing $\gamma_I$ did not significantly impact performance. We note that the effect of increasing $\gamma_E$ is hard to disentangle from the effective increase in the weight of the extrinsic reward in the return. To address this ambiguity we would need to run an extensive hyperparameter sweep of the weights of intrinsic and extrinsic rewards and $\gamma_E$.

### 3.4 RECURRENCE

Montezuma's Revenge is a partially observable environment even though large parts of the game state can be inferred from the screen. For example the number of keys the agent has appears on the screen, but not where they come from, how many keys have been used in the past, or what doors have been opened. To deal with this partial observability, an agent should maintain a state summarizing the past, for example the state of a recurrent policy. Hence it would be natural to hope for better performance from agents with recurrent policies. Contrary to expectations in Figure 6 recurrent policies performed worse than non-recurrent counterparts. We provide an additional experiment confirming this finding in the Appendix (fig. 8). However this finding did not hold true for other games as shown in Section 3.6.

### 3.5 SCALING UP RNN TRAINING

In this section we report experiments showing the effect of increased scale on RNN training. The intrinsic rewards are non-episodic with $\gamma_I = 0.99$, and $\gamma_E = 0.999$.

To hold the rate at which the intrinsic reward decreases over time constant across experiments with different numbers of parallel environments, we downsample the batch size when training the predictor to match the batch size with 32 parallel environments (for full details see Appendix A.4). Larger numbers of environments results in larger batch sizes per update for training the policy, whereas the predictor network batch size remains constant. Since the intrinsic reward disappears over time it is important for the policy to learn to find and exploit these transitory rewards, since they act as stepping-stones to nearby novel states.

Figure 5 shows that agents trained with larger batches of experience collected from more parallel environments obtain higher mean returns after similar numbers of updates. They also achieve better final performance.

We allowed the experiment with 32 parallel environments to run for more time, eventually reaching a mean return of 7,570 after processing 1.6 billion frames over 1.6 million parameter updates. One of these runs visited all 24 rooms, and passed the first level once, achieving a best return of 17,500. The experiment with 1024 parallel environments had mean return of 10,070 at the end of training, and yielded one run with mean return of 14,415.

### 3.6 COMPARISON TO BASELINES

In this section we compare RND to two baselines: PPO without an exploration bonus and an alternative exploration bonus based on forward dynamics error. We evaluate RND's performance on six hard exploration Atari games: Gravitar, Montezuma's Revenge, Pitfall!, Private Eye, Solaris, and Venture. We first compare to the performance of a baseline PPO implementation without intrinsic reward. For RND the intrinsic rewards are non-episodic with $\gamma_I = 0.99$, while $\gamma_E = 0.999$ for both PPO and RND. The results are shown in Figure 7.

In Gravitar we see that RND does not consistently exceed the performance of PPO. However both exceed average human performance with an RNN policy, as well as the previous state of the art. On Montezuma's Revenge and Venture RND significantly outperforms PPO, and exceeds state of the art performance and average human performance. On Pitfall! both algorithms fail to find any positive

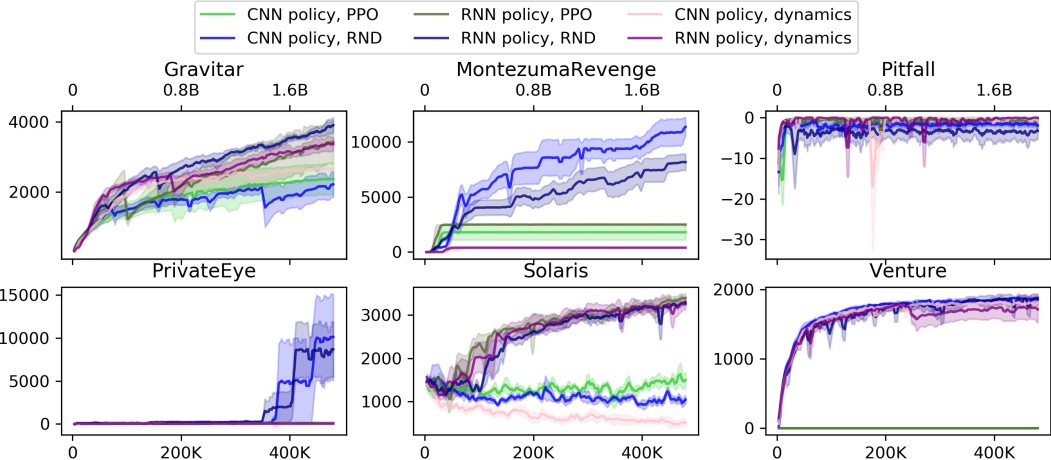

Figure 7: Mean episodic return of RND, dynamics-based exploration method, and PPO with extrinsic reward only on 6 hard exploration Atari games. RND achieves state of the art performance on Gravitar, Montezuma's Revenge, and Venture, significantly outperforming PPO on the latter two. Curves are an average over 3 random seeds. Horizontal axes show numbers of parameter updates at the bottom of the graphs and the numbers of frames at the top.

rewards. This is a typical result for this game, as the extrinsic positive reward is very sparse. On Private Eye RND's performance exceeds that of PPO. On Solaris RND's performance is comparable to that of PPO.

Next we consider an alternative exploration bonus based on forward dynamics error. There are numerous previous works using such a bonus (Schmidhuber, 1991b; Stadie et al., 2015; Achiam & Sastry, 2017; Pathak et al., 2017; Burda et al., 2018). Fortuitously Burda et al. (2018) show that training a forward dynamics model in a random feature space typically works as well as any other feature space when used to create an exploration bonus. This means that we can easily implement an apples to apples comparison and change the loss in RND so the predictor network predicts the random features of the next observation given the current observation and action, while holding fixed all other parts of our method such as dual value heads, non-episodic intrinsic returns, normalization schemes etc. This provides an ablation of the prediction problem defining the exploration bonus, while also being representative of a class of prior work using forward dynamics error. Our expectation was that these methods should be fairly similar except where the dynamics-based agent is able to exploit non-determinism in the environment to get intrinsic reward.

Figure 7 shows that dynamics-based exploration performs significantly worse than RND with the same CNN policy on Montezuma's Revenge, PrivateEye, and Solaris, and performs similarly on Venture, Pitfall, and Gravitar. By analyzing agent's behavior at convergence we notice that in Montezuma's Revenge the agent oscillates between two rooms. This leads to an irreducibly high prediction error, as the non-determinism of sticky actions makes it impossible to know whether, once the agent is close to crossing a room boundary, making one extra step will result in it staying in the same room, or crossing to the next one. This is a manifestation of the 'noisy TV' problem, or aleatoric uncertainty discussed in Section 2.2.1. Similar behavior emerges in PrivateEye and Pitfall!. Table 5 in Appendix A.6 contains further details on the final mean performance of each algorithm.

## 3.7 Qualitative Analysis: Dancing with skulls

By observing the RND agent (goo.gl/DGPC8E), we notice that frequently once it obtains all the extrinsic rewards that it knows how to obtain reliably (as judged by the extrinsic value function), the agent settles into a pattern of behavior where it keeps interacting with potentially dangerous objects. For instance in Montezuma's Revenge the agent jumps back and forth over a moving skull, moves in between laser gates, and gets on and off disappearing bridges. We also observe similar behavior in Pitfall!. It might be related to the very fact that such dangerous states are difficult to achieve, and hence are rarely represented in agent's past experience compared to safer states.

## 4  RELATED WORK

**Exploration.** Count-based exploration bonuses are a natural and effective way to do exploration (Strehl & Littman, 2008) and a lot of work has studied how to tractably generalize count bonuses to large state spaces (Bellemare et al., 2016; Fu et al., 2017; Ostrovski et al., 2018; Tang et al., 2017; Machado et al., 2018; Fox et al., 2018).

Another class of exploration methods rely on errors in predicting dynamics (Schmidhuber, 1991b; Stadie et al., 2015; Achiam & Sastry, 2017; Pathak et al., 2017; Burda et al., 2018). As discussed in Section 2.2, these methods are subject to the 'noisy TV' problem in stochastic or partially-observable environments. This has motivated work on exploration via quantification of uncertainty (Still & Precup, 2012; Houthooft et al., 2016) or prediction improvement measures (Schmidhuber, 1991a; Oudeyer et al., 2007; Lopes et al., 2012; Achiam & Sastry, 2017).

Other methods of exploration include adversarial self-play (Sukhbaatar et al., 2018), maximizing empowerment (Gregor et al., 2017), parameter noise (Plappert et al., 2017; Fortunato et al., 2017), identifying diverse policies (Eysenbach et al., 2018; Achiam et al., 2018), and using ensembles of value functions (Osband et al., 2018; 2016; Chen et al., 2017).

**Montezuma's Revenge.** Early neural-network based reinforcement learning algorithms that were successful on a significant portion of Atari games (Mnih et al., 2015; 2016; Hessel et al., 2017) failed to make meaningful progress on Montezuma's Revenge, not finding a way out of the first room reliably. This is not necessarily a failure of exploration, as even a random agent finds the key in the first room once every few hundred thousand steps, and escapes the first room every few million steps. Indeed, a mean return of about 2,500 can be reliably achieved without special exploration methods (Horgan et al., 2018; Espeholt et al., 2018; Oh et al., 2018).

Combining DQN with a pseudo-count exploration bonus Bellemare et al. (2016) set a new state of the art performance, exploring 15 rooms and getting best return of 6,600. Since then a number of other works have achieved similar performance (O'Donoghue et al., 2017; Ostrovski et al., 2018; Machado et al., 2018; Osband et al., 2018), without exceeding it.

Special access to the underlying RAM state can also be used to improve exploration by using it to hand-craft exploration bonuses (Kulkarni et al., 2016; Tang et al., 2017; Stanton & Clune, 2018). Even with such access previous work achieves performance inferior to average human performance.

Expert demonstrations can be used effectively to simplify the exploration problem in Montezuma's Revenge, and a number of works (Salimans & Chen, 2018; Pohlen et al., 2018; Aytar et al., 2018; Garmulewicz et al., 2018) have achieved performance comparable to or better than that of human experts. Learning from expert demonstrations benefits from the game's determinism. The suggested training method (Machado et al., 2017) to prevent an agent from simply memorizing the correct sequence of actions is to use sticky actions (i.e. randomly repeating previous action) has not been used in these works. In this work we use sticky actions and thus don't rely on determinism.

**Random features.** Features of randomly initialized neural networks have been extensively studied in the context of supervised learning (Rahimi & Recht, 2008; Saxe et al., 2011; Jarrett et al., 2009; Yang et al., 2015). More recently they have been used in the context of exploration (Osband et al., 2018; Burda et al., 2018). The work Osband et al. (2018) provides motivation for random network distillation as discussed in Section 2.2.

**Vectorized value functions.** Pong et al. (2018) find that a vectorized value function (with coordinates corresponding to additive factors of the reward) improves their method. Bellemare et al. (2017) parametrize the value as a linear combination of value heads that estimate probabilities of discretized returns. However the Bellman backup equation used there is not itself vectorized. More broadly, the issue of how to approach optimizing multiple objectives is an important topic in reinforcement learning, see (Roijers et al., 2013).

## 5  DISCUSSION

This paper introduced an exploration method based on random network distillation and experimentally showed that the method is capable of performing directed exploration on several Atari games with very sparse rewards. These experiments suggest that progress on hard exploration games is possible with relatively simple generic methods, especially when applied at scale. They also suggest that

methods that are able to treat the stream of intrinsic rewards separately from the stream of extrinsic rewards (for instance by having separate value heads) can benefit from such flexibility.

We find that the RND exploration bonus is sufficient to deal with local exploration, i.e. exploring the consequences of short-term decisions, like whether to interact with a particular object, or avoid it. However global exploration that involves coordinated decisions over long time horizons is beyond the reach of our method.

To solve the first level of Montezuma's Revenge, the agent must enter a room locked behind two doors. There are four keys and six doors spread throughout the level. Any of the four keys can open any of the six doors, but are consumed in the process. To open the final two doors the agent must therefore forego opening two of the doors that are easier to find and that would immediately reward it for opening them.

To incentivize this behavior the agent should receive enough intrinsic reward for saving the keys to balance the loss of extrinsic reward from using them early on. From our analysis of the RND agent's behavior, it does not get a large enough incentive to try this strategy, and only stumbles upon it rarely.

Solving this and similar problems that require high level exploration is an important direction for future work.

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

## A  APPENDIX

### A.1  ADDITIONAL METHODOLOGICAL DETAILS

#### A.1.1  REWARD AND OBSERVATION NORMALIZATION

One issue with using prediction error as an exploration bonus is that the scale of the reward can vary greatly between different environments and at different points in time, making it difficult to choose hyperparameters that work in all settings. In order to keep the rewards on a consistent scale we normalized the intrinsic reward by dividing it by a running estimate of the standard deviations of the intrinsic returns.

Observation normalization is often important in deep learning but it is crucial when using a random neural network as a target, since the parameters are frozen and hence cannot adjust to the scale of different datasets. Lack of normalization can result in the variance of the embedding being extremely low and carrying little information about the inputs. To address this issue we use an observation normalization scheme often used in continuous control problems whereby we whiten each dimension by subtracting the running mean and then dividing by the running standard deviation. We then clip the normalized observations to be between -5 and 5. We initialize the normalization parameters by stepping a random agent in the environment for a small number of steps before beginning optimization. We use the same observation normalization for both predictor and target networks but not the policy network.

#### A.1.2  REINFORCEMENT LEARNING ALGORITHM

An exploration bonus can be used with any RL algorithm by modifying the rewards used to train the model (i.e., $r_t = i_t + e_t$). We combine our proposed exploration bonus with a baseline reinforcement learning algorithm PPO (Schulman et al., 2017). PPO is a policy gradient method that we have found to require little tuning for good performance. For algorithmic details see Algorithm 1.

### A.2  RND PSEUDO-CODE

Algorithm 1 gives an overall picture of the RND method. Exact details of the method can be found in the code accompanying this paper (goo.gl/DGPC8E).

---

**Algorithm 1** RND pseudo-code

---

$N \leftarrow$ number of rollouts
$N_{\text{opt}} \leftarrow$ number of optimization steps
$K \leftarrow$ length of rollout
$M \leftarrow$ number of initial steps for initializing observation normalization
$t = 0$
Sample state $s_0 \sim p_0(s_0)$
**for** $m = 1$ **to** $M$ **do**
    sample $a_t \sim \text{Uniform}(a_t)$
    sample $s_{t+1} \sim p(s_{t+1}|s_t, a_t)$
    Update observation normalization parameters using $s_{t+1}$
    t += 1
**end for**
**for** $i = 1$ **to** $N$ **do**
    **for** $j = 1$ **to** $K$ **do**
        sample $a_t \sim \pi(a_t|s_t)$
        sample $s_{t+1}, e_t \sim p(s_{t+1}, e_t|s_t, a_t)$
        calculate intrinsic reward $i_t = \|\hat{f}(s_{t+1}) - f(s_{t+1})\|^2$
        add $s_t, s_{t+1}, a_t, e_t, i_t$ to optimization batch $B_i$
        Update running estimate of reward standard deviation using $i_t$
        t += 1
    **end for**
    Normalize the intrinsic rewards contained in $B_i$
    Calculate returns $R_{I,i}$ and advantages $A_{I,i}$ for intrinsic reward
    Calculate returns $R_{E,i}$ and advantages $A_{E,i}$ for extrinsic reward
    Calculate combined advantages $A_i = A_{I,i} + A_{E,i}$
    Update observation normalization parameters using $B_i$
    **for** $j = 1$ **to** $N_{\text{opt}}$ **do**
        optimize $\theta_\pi$ wrt PPO loss on batch $B_i, R_i, A_i$ using Adam
        optimize $\theta_{\hat{f}}$ wrt distillation loss on $B_i$ using Adam
    **end for**
**end for**

---

## A.3 PREPROCESSING DETAILS

Table 1 contains details of how we preprocessed the environment for our experiments. We followed the recommendations in Machado et al. (2017) in using sticky actions in order to make the environments non-deterministic so that memorization of action sequences is not possible. In Table 2 we show additional preprocessing details for the policy and value networks. In Table 3 we show additional preprocessing details for the predictor and target networks.

| Hyperparameter | Value |
|---|---|
| Grey-scaling | True |
| Observation downsampling | (84,84) |
| Extrinsic reward clipping | $[-1, 1]$ |
| Intrinsic reward clipping | False |
| Max frames per episode | 18K |
| Terminal on loss of life | False |
| Max and skip frames | 4 |
| Random starts | False |
| Sticky action probability | 0.25 |

Table 1: Preprocessing details for the environments for all experiments.

| Hyperparameter | Value | | Hyperparameter | Value |
|---|---|---|---|---|
| Frames stacked | 4 | | Frames stacked | 1 |
| Observation normalization | $x \mapsto x/255$ | | Observation normalization | $x \mapsto \text{CLIP}\left((x - \mu)/\sigma, [-5, 5]\right)$ |

Table 2: Preprocessing details for policy and value network for all experiments.

Table 3: Preprocessing details for target and predictor networks for all experiments.

### A.4 PPO AND RND HYPERPARAMETERS

In Table 4 the hyperparameters for the PPO RL algorithm along with any additional hyperparameters used for RND are shown. Complete details for how these hyperparameters are used can be found in the code accompanying this paper.

| Hyperparameter | Value |
|---|---|
| Rollout length | 128 |
| Total number of rollouts per environment | 30K |
| Number of minibatches | 4 |
| Number of optimization epochs | 4 |
| Coefficient of extrinsic reward | 2 |
| Coefficient of intrinsic reward | 1 |
| Number of parallel environments | 128 |
| Learning rate | 0.0001 |
| Optimization algorithm | Adam (Kingma & Ba (2015)) |
| $\lambda$ (Schulman et al., 2017) | 0.95 |
| Entropy coefficient | 0.001 |
| Proportion of experience used for training predictor | 0.25 |
| $\gamma_E$ | 0.999 |
| $\gamma_I$ | 0.99 |
| Clip range | $[0.9, 1.1]$ |

Table 4: Default hyperparameters for PPO and RND algorithms for experiments where applicable. Any differences to these defaults are detailed in the main text.

Initial preliminary experiments with RND were run with only 32 parallel environments. We expected that increasing the number of parallel environments would improve performance by allowing the policy to adapt more quickly to transient intrinsic rewards. This effect could have been mitigated however if the predictor network also learned more quickly. To avoid this situation when scaling up from 32 to 128 environments we kept the effective batch size for the predictor network the same by randomly dropping out elements of the batch with keep probability 0.25. Similarly in our experiments with 256 and 1,024 environments we dropped experience for the predictor with respective probabilities 0.125 and 0.03125.

### A.5 ARCHITECTURES

In this paper we use two policy architectures: an RNN and a CNN. Both contain convolutional encoders identical of those in the standard architecture from (Mnih et al., 2015). The RNN architecture additionally contains GRU (Cho et al., 2014) cells to capture longer contexts. The architectures of the target and predictor networks also have convolutional encoders identical to the ones in (Mnih et al., 2015) followed by dense layers. Exact details are given in the code accompanying this paper (goo.gl/DGPC8E).

### A.6 ADDITIONAL EXPERIMENTAL RESULTS

Figure 8 compares the performance of a recurrent policy to a CNN policy with access to only the last 16 most recent frames with a matched number of parameters. The intrinsic rewards were non-episodic

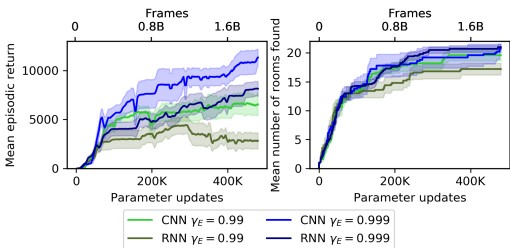 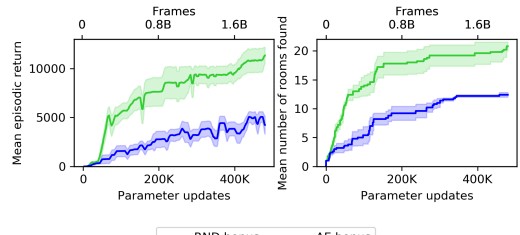

Figure 8: Comparison of recurrent and non-recurrent policies with the same number of parameters with extrinsic reward discount factors $\gamma_E \in \{0.99, 0.999\}$. Similar to the results in Figure 4, higher discount factors lead to better performance. Contrary to our expectations recurrent policies perform worse than non-recurrent counterparts. Curves are an average over 5 random seeds.

Figure 9: Comparison of RND with a CNN policy with $\gamma_I = 0.99$ and $\gamma_E = 0.999$ with an exploration defined by the reconstruction error of an autoencoder, holding all other choices constant (e.g. using dual value, treating intrinsic return as non-episodic etc). The performance of the autoencoder-based agent is worse than that of RND, but exceeds that of baseline PPO. Curves are an average over 5 random seeds.

with $\gamma_I = 0.99$. Here again we see that the CNN policy consistently outperforms the RNN policy on Montezuma's Revenge.

Figure 9 compares the performance of RND with an identical algorithm, but with the exploration bonus defined as the reconstruction error of an autoencoder. The autoencoding task is similar in nature to the random network distillation, as it also obviates the second (though not necessarily the third) sources of prediction error from section 2.2.1. The experiment shows that the autoencoding task can also be successfully used for exploration.

In Table 5 we see more details of the experiments in Section 3.6. There the final training performance for each algorithm is listed, alongside the state of the art from previous work and average human performance.

|  | Gravitar | Montezuma's Revenge | Pitfall! | PrivateEye | Solaris | Venture |
|---|---|---|---|---|---|---|
| RND RNN | **3,906** | 8,152 | -3 | 8,666 | 3,282 | **1,859** |
| PPO RNN | 3,426 | 2,497 | 0 | 105 | 3,387 | 0 |
| RND CNN | 2,217 | **11,347** | -2 | 10,117 | 1,050 | **1,878** |
| DYN CNN | 2,654 | 400 | -1 | 31 | 515 | **1,807** |
| PPO CNN | 2,370 | 1,797 | 0 | 100 | 1,495 | 0 |
| SOTA | 2,209[1] | 3,700[2] | **0** | 15,806[2] | 12,380[1] | 1,813[3] |
| Avg. Human | 3,351 | 4,753 | 6,464 | 69,571 | 12,327 | 1,188 |

Table 5: Comparison to baselines results. Final mean performance for various methods. State of the art results taken from: [1] (Fortunato et al., 2017) [2] (Bellemare et al., 2016) [3] (Horgan et al., 2018)

## A.7 ADDITIONAL EXPERIMENTAL DETAILS

In Table 6 we show the number of seeds used for each experiment, indexed by figure.

| Figure number | Number of seeds |
|:---:|:---:|
| 1 | NA |
| 2 | 10 |
| 3 | 5 |
| 4 | 5 |
| 5 | 10 |
| 6 | 5 |
| 7 | 3 |
| 8 | 5 |
| 9 | 5 |

Table 6: The numbers of seeds run for each experiment is shown in the table. The results of each seed are then averaged to provide a mean curve in each figure, and the standard error is used make the shaded region surrounding each curve.

