# OpenReview forum: "Exploration by random network distillation"
_ICLR.cc/2019/Conference_

### Official Review · AnonReviewer2 · 2018-11-02
**A simple yet surprisingly effective take on intrinsic motivation for exploration in sparse reward RL tasks**

**Rating:** 7
**Confidence:** 4

**Review:**

The algorithm proposed in this paper consists in driving exploration in RL through an intrinsic reward, computed as the prediction error of a neural network whose target is the output of a randomly initialized network (with the state reached by the agent as input). The intuition is that rarely seen states will have a large prediction error, thus encouraging the agent to visit them (until they have been seen often enough that the error goes down). Among potential benefits of this method, compared to previously proposed intrinsic curiosity techniques for RL, are its simplicity and its robustness to environment stochasticity. Extensive experiments on the Atari game Montezuma’s Revenge investigate several variants of this idea (combined with PPO), with the best results significantly outperforming the current state-of-the-art. Other results on five other hard exploration Atari games show competitive performance as well.

The proposed technique definitely exhibits impressive performance on some tasks, in spite of its simplicity. Despite lacking theoretical grounding, I believe such results should be quite interesting to the RL research & applied community, as a novel and easy way to encourage exploration in sparse rewards tasks. I also really appreciate that the authors have included “negative” results contradicting their expectations, and are sharing their code: this is the kind of openness that in my opinion should be highly encouraged.

The paper is overall well written and easy to follow, except (from my point of view) section 2.2.2, which I found rather confusing and not very convincing. First, eq. 1 is a bit surprising since one expects the posterior to be in the same family of functions, i.e. of the form f_theta rather than f_theta + f_theta*. After a (very superficial) look at Osband et al (2018) I see that this particular lemma holds for linear functions, and the extension to nonlinear function approximation seems to be essentially based on intuition. Then the sentence “the optimization problem (...) is equivalent to distilling a randomly drawn function from the prior” ignores the sign mismatch (we are actually distilling the opposite of f_theta*, though I agree it can still make sense with a symmetric prior around 0, which is not mentioned). Finally, the reasoning to reach the conclusion “the distillation error could be seen as a quantification of uncertainty in predicting the constant zero function” seems somewhat unconvincing to me, considering the significant differences compared to Osband et al (2018), in particular: sharing weights among models in the ensemble, ignoring the specific regularization term R(theta), and not adding noise to the training data. As a result I find this link rather weak and I would appreciate if this section could be improved (at the very least with a better explanation of its limitations)

Among the various findings from experiments, one puzzled me in particular: the striking difference between episodic and non episodic intrinsic rewards in Fig. 3. I think this would have deserved a more thorough empirical investigation than the intuitive explanation from 2.3 (e.g. by checking whether the agent trained with non episodic rewards was indeed taking more risks and thus dying more often). What I find particularly surprising is that the beginning of the game should not yield much intrinsic reward relatively fast, since it should be the part the agent sees most often initially. As a result, I would expect that getting zero reward when dying (episodic rewards) should not be much different from getting future (small and discounted) intrinsic rewards, unless maybe early in training. What am I missing here?

I also have some comments regarding a couple of other findings and associated hypotheses:
- Section 3.3 shows some surprising results when varying discount factors (“This is at odds with the results in Figure 3 where increasing gamma_I did not significantly impact performance”). I wonder however to which extent these may be caused by the difference in the scale of discounted returns: for instance increasing gamma_I from 0.99 to 0.999 will (roughly) multiply V_I by 10, giving it more weight in the sum V = V_E + V_I. A fair comparison would either rescale V_I accordingly , or use a weighted sum and optimize the weights (the hyper-parameters table in the Appendix suggests that weights were actually used, but they are not mentioned in the main text and it is not clear how they were chosen).
- 3.7 shows an interesting behavior (“dancing with skulls”). The authors hypothetize it may be due to the inherent danger of such behavior. But could it be also (and possibly more) related to the fact the skulls are moving? (which leads to many varied different states, that the predictor network will take time to learn perfectly).

Here are a few more questions for the authors regarding specific details:
1. In 3.1, “The best return achieved by 4 out 5 runs of this setting was 6,700.” What does this mean?
2. In 3.5 a downsampling scheme is used to keep the training speed of the predictor network constant when increasing the number of actors. This raises the question of the impact of this training speed on the results, which is not investigated in the current experiments: do hyper-parameters influencing the predictor’s training speed (e.g. downsampling ratio, learning rate) need to be very carefully tuned, or are results robust across a wide range of speeds?
3. In A.5 there is mention of “a CNN policy with access to only the last 16 most recent frames”: does that mean the number of “frames stacked” (Table 2) was increased from 4 to 16? If so, why? (it is not clear to me what we learn compared to Fig. 4)
4. Your technique implicitly relies on the assumption that the predictor network’s weights will never be exactly the same as the target network’s (as otherwise nothing will be novel anymore, regardless of the states being visited). Do you foresee potential issues with this, and if yes do you have any idea to solve them? (a short discussion in the paper on this topic would be good as well)

And finally some suggestions for small improvements:
- Please try to find another name than “target” network since it is already widely used in the deep RL literature for something completely different (suggestions: “random”, “distillation”, “feature”, “reference”)
- In 2.1 (last paragraph) there are various papers cited regarding forward or inverse dynamics, but several of them contain both, while the way they are cited suggests they deal only with one. Just moving “and inverse dynamics” before the full list of citations would fix it.
- In first paragraph of section 3 please mention that the algorithm is based on PPO
- In Fig. 3 the x axis seems to be missing a multiplication by 1K (?)
- At end of 3.2, “having two value heads is necessary for combining reward streams with different characteristics”, please specify what are these characteristics.
- On p. 7, last paragraph: please (briefly) explain how the “random features” are computed
- The reference Ostrovski et al appears twice
- In Alg. 1, “Update reward normalization parameters using it”: the “s” in parameters can be misleading, suggesting that both mean and standard deviation are used for normalization => explicitly saying “Update running standard deviation” would avoid such confusion (or say it on the “Normalize” step below)
- Alg. 1 is not very clear on how returns and advantages are computed (and the corresponding code is not super easy to read). It also seems to be missing the update of the critic V.
- Alg. 1 mentions “number of optimization steps” while Table 4 says “Number of optimization epochs”: I guess they are the same, so they should probably have the same name
- After reading the paper, I felt like one learning was that CNN models worked better than RNN ones. However Table 5 shows that this can vary between games (ex: RND RNN outperforms RND CNN on Gravitar and Solaris) and/or algorithms (ex: PPO RNN outperforms PPO CNN on 3 games). I think the main text should at least point to this table when mentioning the superiority of the CNN.
- In the “Related work” section there is a very short paragraph about “vectorized value functions”. It seems to be overlooking the whole field of multi-objective reinforcement learning. Maybe you could cite a related survey paper like “A Survey of Multi-Objective Sequential Decision-Making”.
- The paper’s title and the OpenReview submission name should probably match

Update following author and reviewer discussion: I agree with others regarding the weakness of the empirical comparison to pseudo-counts in particular, but still believe that the paper deserves to be accepted due to the fact that (1) some of the results are really good, and (2) this is a simple original idea that has the potential to drive further advances (hopefully addressing the empirical and theoretical limitations of the current work)

---

> ### Author Response · Authors · 2018-11-21
> **Reply to reviewer 2**
>
> Thank you for your detailed review, we have updated the paper to accommodate your suggested improvements.
>
> We have rewritten section 2.2.2 to emphasize that the generalization to non-linear functions is heuristic, and added details to the connection with lemma 3 from Osband et al. We agree that there are important differences but we suspect that a similar mechanism may underlie the performance of both approaches and wanted to bring the connection to the reader’s attention.
>
> "the striking difference between episodic and non episodic intrinsic rewards"
>
> In our experience episodic intrinsic only agents typically became trapped in the first room. We believe that this is because early on in training the effective penalty for non-episodic agents is low, allowing to explore the immediate surroundings of the starting states. Further exploration is contingent upon successful exploration of the immediate surroundings and so the softening of the game-over penalty early in training can have a large compounded effect. You are correct that this deserves further investigation.
>
> “Section 3.3 shows some surprising results when varying discount factors (“This is at odds with the results in Figure 3 where increasing gamma_I did not significantly impact performance”). I wonder however to which extent these may be caused by the difference in the scale of discounted returns: for instance increasing gamma_I from 0.99 to 0.999 will (roughly) multiply V_I by 10, giving it more weight in the sum V = V_E + V_I. A fair comparison would either rescale V_I accordingly , or use a weighted sum and optimize the weights (the hyper-parameters table in the Appendix suggests that weights were actually used, but they are not mentioned in the main text and it is not clear how they were chosen).”
>
> We agree that it’s possible the effect that you mention could be contributing to the difference in results and we have updated the text to reflect this possibility. The heuristic you mention of multiplying by 10 might be reasonable in some cases, but in general we would have to perform a hyperparameter sweep per game to get the right value (since it will depend significantly on the sparsity of the reward). We tried gamma 0.999 because prior works using learning from demonstrations on Montezuma’s Revenge had suggested that this was effective for Montezuma’s Revenge where a typical episode of a well-performing agent lasts between a 1000 and 4000 steps.
>
> “3.7 shows an interesting behavior (“dancing with skulls”) ... But could it be also (and possibly more) related to the fact the skulls are moving? (which leads to many varied different states, that the predictor network will take time to learn perfectly).”
>
> It’s likely that the fact the object is moving is an important factor, but that doesn’t explain why the agent prefers to dance with the skull rather than observe it from a safe position.
>
> “1. In 3.1, “The best return achieved by 4 out 5 runs of this setting was 6,700.” What does this mean?”
> We mean that the high score of the training run was 6700 is 4 out of 5 random seeds. The maximum score achieved is a useful metric in the pure exploration setting.
>
> “2. In 3.5 a downsampling scheme is used to keep the training speed of the predictor network constant when increasing the number of actors. ... do hyper-parameters influencing the predictor’s training speed (e.g. downsampling ratio, learning rate) need to be very carefully tuned, or are results robust across a wide range of speeds?”
>
> In our preliminary experiments with 32 actors we observed good performance with no downsampling and default learning rate. When we increased the number of actors we simply increased the downsampling rate to match the effective batch size with 32 actors to avoid an additional confounding factor. Likely performance can be improved by tuning this parameter but we did not run many experiments with alternative setups.
>
> “3. In A.5 there is mention of “a CNN policy with access to only the last 16 most recent frames”: does that mean the number of “frames stacked” (Table 2) was increased from 4 to 16? If so, why? (it is not clear to me what we learn compared to Fig. 4)”
>
> By 16 frames we refer to the standard setup for RL in Atari games. Each observation is a max over 4 frames, and the agent sees a stack of 4 such observations, giving a total context of 16 frames.
>
> “4. Your technique implicitly relies on the assumption that the predictor network’s weights will never be exactly the same as the target network’s (as otherwise nothing will be novel anymore, regardless of the states being visited). Do you foresee potential issues with this, and if yes do you have any idea to solve them?”
>
> If the distribution of states is wide enough, the optimization might drive the networks to approach each other asymptotically. However we have not observed this happening in any of our experiments - the reward signal remained meaningful even after tens of billions of frames of processed experience.

---

> > ### Comment · AnonReviewer2 · 2018-11-23
> > **Re: Reply to reviewer 2**
> >
> > Thanks for the replies & revision! A few follow-ups:
> >
> > "It’s likely that the fact the object is moving is an important factor, but that doesn’t explain why the agent prefers to dance with the skull rather than observe it from a safe position."
> >
> > Good point, though a counter-argument could be that the combination of both the moving agent *and* skulls in a small region of the screen leads to a more challenging modeling task than having them far from each other. An experiment with non-deadly skulls might settle the argument but it's probably not worth it :)
> >
> > "We mean that the high score of the training run was 6700 is 4 out of 5 random seeds"
> >
> > Ok thanks (with "in" instead of "is"). I think it may be worth clarifying in the paper, basically I got confused because I didn't imagine that 6700 was a score shared by 4 different runs, so I thought it was a best over the seeds, but then the "4 out of 5" made no sense (side note on a typo: "of" is missing in the paper). You could write for instance: "During training, 4 out of the 5 agents (trained with different seeds) reached the same peak score of 6700[, while the 5th one reached at most XXX]" (add the high score of the last agent if available).

---

### Official Review · AnonReviewer4 · 2018-11-08
**Efficient and simple to implement exploration strategy for RL**

**Rating:** 10
**Confidence:** 4

**Review:**

The paper presents a simple but remarkably efficient exploration strategy obtaining state-of-art results in a well known hard-exploration problem (Montezuma's Revenge). The idea consists of several parts:
1. The authors suggested distilling a fixed randomly initialized network into another randomly initialized trained network in order to use prediction errors as pseudo-rewards. The authors claim that distillation error is a proxy for visit-counts and experimentally demonstrate this idea on MNIST dataset.
2. The authors suggested using two separate value heads to evaluate expected rewards and expected pseudo-rewards with different time horizons (discount factors) under the same policy.

The paper is overall well written and easy to read. As far as I can tell, the use of a distillation error as an exploration reward is novel. Relative efficiency of the method compared to its simplicity should interest most people working in RL.

The main problem which I see is the presentation of learning curves as a function of training steps rather than acting steps. While I acknowledge that the achievement of state-of-art asymptotic performance is valuable on its own, presenting results as a function of acting steps (rather than parameter update steps) may better show data and exploration efficiency. This would also facilitate comparisons with other RL algorithms which may have different architectures (for example, multiple networks updated at different frequencies).

I liked the idea to use two value heads to evaluate intrinsic and extrinsic values with different discounts. Still, as both heads share a common 'trunk' network, they will inevitably affect each other. For example, scaling the pseudo-rewards by 10 and scaling the pseudo-reward value function by 0.1 to produce the same summed value function may lead to a different training dynamics due to the influence of intrinsic value head onto the extrinsic one. Are the results sensitive to this effect? Also, how are the results sensitive to the scale of pseudo-rewards? What would happen if they were simply multiplied or divided by 10?

Also, what was the distributed training setting that you used to train your agent? Were the actors running on a single machine or on multiple machines? Was a single trainer running on a single machine training network on batched observations or was training distributed in some way? The reason why I am asking this is that as a distillation error fundamentally depends on its training dynamics, I would not be surprised if the results could be affected by the training setting. For example, if the network was trained in a distributed setting, asynchronous updates could introduce implicit momentum and thus may cause a pseudo reward to oscillate. While I do not think that is a fundamental problem with the work either way, it would be nice to know a few more details for future reproducibility.

Other minor comments:
Figure 2. It would be nice to see if both x and y axes was plotted in log scale in order to visualize any power-law (if one exists) between samples and MSE.
Figure 3. I would prefer 'x' axis to be in the number of steps.
Figure 4. Again, performance between different actor-configurations would be easier to see if x axis was a total number of steps, as it would be easier to see if the curves overlap and the method scales linearly with the number of actors.

---

> ### Author Response · Authors · 2018-11-21
> **Reply to Reviewer 4**
>
> Thank you for your comments, we are glad that you enjoyed this work.
>
> “The main problem which I see is the presentation of learning curves as a function of training steps rather than acting steps”
>
> We have updated the paper to include the number of frames in the figures.
>
> “How are the results sensitive to the scale of pseudo-rewards? What would happen if they were simply multiplied or divided by 10?”
>
> We use a reward normalization scheme that brings the intrinsic reward to a predictable range. If we were to up-weigh the coefficient of the intrinsic reward in the value calculation, the results would indeed change. We found the method to be relatively stable to changes in hyper-parameters, but it is of course possible to break it by large changes.
>
> “Also, what was the distributed training setting that you used to train your agent? Were the actors running on a single machine or on multiple machines?”
>
> The majority of the experiments were done on a single gpu, and the others used 8 gpus on a single machine (for the larger scale experiments, or to complete the smaller scale experiments more quickly as we approached the deadline). The parallelization was handled using MPI and was completely synchronous.
>
> “Figure 2. It would be nice to see if both x and y axes was plotted in log scale in order to visualize any power-law (if one exists) between samples and MSE.”
>
> We have included a log-log scale plot at the link below
> https://pasteboard.co/HO77MWw.png

---

### Official Review · AnonReviewer5 · 2018-11-09
**Clear writing, strong results, sensible algorithm, good paper**

**Rating:** 9
**Confidence:** 5

**Review:**

This paper presents an approach to exploration in RL via random network distillation.
The agent generates a random neural network, and adds an "intrinsic reward" based on the regression error of this random function.
The main evidence for its efficacy comes from evaluation on Atari games, particularly Montezuma's revenge, where it attains state of the art results.

There are several things to like about this paper:

- The writing is clear and well thought out.
- The actual algorithm is sensible, simple, intuitive and clearly effective.
- The results are significant: this is really a "step change" compared to previous Montezuma results.
- This work takes the well-known "exploration bonus" approach, combines it with some of the observations of (Osband et al) and simplifies the treatment... so in some ways it's quite standard... but there are several new insights:
  + Focus on normalization schemes for "randomized prior function"
  + Bootstrapping "intrinsic reward" over episode boundaries
  + Incorporating large-scale policy-based algorithms

To help improve the paper, I will highlight some potential issues:

- For a paper on exploration, it does not make sense to present results in terms of "parameter updates". This should instead be presented in terms of actor/environment steps. This is something that happens consistently across the paper. If you want to show that "many actors makes it better" then you can divide this by #actors... so that the curves still functionally look the same. This is an easy thing to change... but I think it's important to do this!
- Like other "count-based" methods, this exploration bonus is not linked to the task. As such, you have to get "lucky" that you do the right kind of generalization from the "random network". I think that you should mention this issue, potentially in your section 2. That is not to say that this is therefore a bad method, but particularly with reference to (Osband et al 2018) this approach does not address their observation from Section 2.4 of that paper... you don't necessarily get the "right" type of generalization from this random network (that has nothing to do with the task). You could then point out that, empirically, using a random convnet seems to do just fine in Atari! ;D
- The whole section about "pure exploration" is somewhat interesting, but you shouldn't assess that performance in terms of "reward"... because that is just a peculiarity of these games... we could easily imagine a game where "pure exploration" gives a huge negative reward... but that wouldn't mean that it was bad at pure exploration! Therefore, how can you justify the quality of pure exploration by reference to the "best return".
- Although the paper is definitely good, and I've already outlined several truly novel additions from this paper, on another level the actual intellectual contribution of this paper is perhaps not *as* large as it may seem from the Abstract or associated OpenAI publicity/blog posts https://blog.openai.com/reinforcement-learning-with-prediction-based-rewards/
  + This paper is about adding an "exploration bonus" to RL rewards (this goes back at least to Kearns+Singh 2002)
  + The form of this bonus comes from prediction error on a random function
  + I have some concerns on the process of "anonymous" reviews in this "blog+tweet" setting

Overall, I like the paper a lot, I think it must be accepted and also it's right at the top of ICLR best papers!
The writing is good, the results are good, the algorithm is good and I think it will have impact.
The main missing piece is a clear discussion of any of the algorithms potential weaknesses - is this the final solution to exploration? What do you think about the issues of generalization? How would this perform in a linear system? What if the basis functions are not aligned?
It's not that the algorithm needs to address all of these things to be a good algorithm, but the paper should try to do a better job about highlighting any potential missing pieces - particularly when the results are so impressive.

---

> ### Author Response · Authors · 2018-11-21
> **Reply to Reviewer 5**
>
> Thank you for your helpful review and we are glad that you like the paper.
>
> “- For a paper on exploration, it does not make sense to present results in terms of "parameter updates". This should instead be presented in terms of actor/environment steps. ”
>
> We have updated the paper to include the number of frames used.
>
> “Like other "count-based" methods, this exploration bonus is not linked to the task. As such, you have to get "lucky" that you do the right kind of generalization from the "random network".”
>
> Our bonus is indeed not linked to the task in the way that Osband et al’s is. However since the problems we are most interested are those where the extrinsic reward is very sparse, these bonuses will behave similarly. Most uncertainty about a sparse reward function (and hence about the optimal policy) comes from finding instances of positive reward. However in dense reward settings the RND bonus might cause the agent to over-explore leading to slower learning in some situations, but we have not investigated this effect.
>
> “The whole section about "pure exploration" is somewhat interesting, but you shouldn't assess that performance in terms of "reward"... because that is just a peculiarity of these games... we could easily imagine a game where "pure exploration" gives a huge negative reward... but that wouldn't mean that it was bad at pure exploration! Therefore, how can you justify the quality of pure exploration by reference to the "best return".”
>
> We agree that in general the extrinsic reward is not an ideal measure of pure exploration. For this reason we also included the number of rooms discovered by the agent which we feel is a much better metric.
>
> “The main missing piece is a clear discussion of any of the algorithms potential weaknesses - is this the final solution to exploration? What do you think about the issues of generalization? How would this perform in a linear system? What if the basis functions are not aligned?”
>
> We would definitely not want to give the impression that this is a final solution for exploration. We believe it is a scalable alternative to count-like bonuses without some of the issues plaguing dynamics-based prediction bonuses. Similar to other methods (for example count-based), our method could be improved by providing a fine-grained way to control generalization to new states and the injection of prior information. A method combining the theoretical grounding of information-gain style approaches with the computational tractability of heuristic methods such as RND is highly desirable but currently out of reach.

---

> > ### Comment · AnonReviewer5 · 2018-11-21
> > **Response**
> >
> > Thanks for your response - I find it all very reasonable.
> >
> > As far as I'm concerned: the more you can include these sentiments in the paper the better.
> > This might mean adding a footnote or section in an appendix.
> > I also think it would be valuable to highlight some of these shortcomings / open challenges in your conclusion - it does not take any shine away from these impressive results, but can help to shape the direction of future research!

---

### Official Review · AnonReviewer1 · 2018-11-15
**Promising method but poor evaluation and presentation**

**Rating:** 4
**Confidence:** 4

**Review:**

My apologies for posting late, I was seriously injured around the reviewer deadline.

---------------------------------

The authors propose "random network distillation," a method that adds an additional reward based on a proxy for "exploration" to the RL task at hand. The method works by including an extra term in the reward during training. The term is calculated as follows. A randomly initialized network is created during rollout generation. Another network is initialized as well, and during rollouts is trained to predict the output of the randomly initialized network applied to the states. The agent then uses a measure of the prediction loss as an intrinsic reward. These rewards are then included as part of the trajectory, and are predicted separately for training purposes.

The authors find that when you combine these intrinsic rewards with agents trained at extremely large scale (~2 billion frames per training run!) it is possible to perform very well on Montezuma's revenge and other sparse reward tasks.

Overall, the paper has great potential - it presents the first algorithm to solve a challenging sparse reward RL task. However, while the method itself is promising, the weak baselines (in particular, the lack of evidence disentangling the benefits of larger scale / more frames vs the benefits of the proposed method) and unclear presentation make me unable to yet recommend the paper for acceptance.

Positive:
 - The work reaches the state-of-the-art on several sparse reward tasks, most notably Montezumas revenge
 - On Montezumas revenge, the method is able to pass through the first level, and explore the vast majority of rooms.
 - The reward mechanism seems to be novel

Negative:
 - All previous work used more than an order of magnitude more frames in training. From the experiments given, it is impossible to distinguish the impact of RND vs larger scale training
 - The baselines are not very strong: The forward dynamics baseline does significantly worse on Montezumas revenge than the previous results in Ostrovski et al and Bellemare et al, even using more than an order of magnitude more frames.
 - Important experimental details lack adequate descriptions
 - Tables and figures are not written with adequate details

Details of negative feedback:

Major:
-------------
Unclear baselines and questionable improvement on SOTA:

 - Previous work (the neural density functions of Ostrovski et al or the CTS scheme of Bellemare et al.) used significantly fewer (~100 million and ~150 million respectively vs ~2 billion) frames of experience in solving Montezumas Revenge, which makes this method’s benefit somewhat incomparable to previous methods given the sampling regime it operates in.
 - It is important to disentangle the impacts of:

   (1) Using many more (an order of magnitude) frames than previous methods
   (2) The presented RND bonus method

  and it is impossible to separate these without further extensive experimentation with previous methods. The main claim of the paper is that the RND bonus is a better method for solving hard exploration games; this needs to be shown through a rigorous comparison.
 - The fact that the forward dynamics does worse than vanilla PPO (and the previous results in Ostrovski et al and Bellemare et al) on Montezuma's revenge brings the strength of the used baseline into question


Overall, the experimental details are greatly lacking:

 - The way that the value function is trained (i.e. the objective function) is never explained in the paper. The value function in PPO is typically (according to the baselines repository) trained at each step to fit (GAE advantage + previous value), but in the paper this is not elaborated on.
 - If this is indeed the case, then the statement that the extrinsic value function fits a stationary distribution on page 5 should be fixed.
 - In Table 4 the $\lambda$ hyperparameter is listed, but is not described at all in the paper. I am guessing that it is the corresponding GAE hyperparameter, but I am not sure as the GAE method is never written about or cited throughout the paper.
 - The paper is not written in a way that is accessible to people that do not closely follow the line of work on sparse rewards. For example, though it is possible to infer, the paper never explicitly defines the intrinsic reward $i_t$ in the main paper text. The exact mechanism through which the "forward dynamics" baseline is never given.


Tables and figures do not give sufficient detail to know what they are describing:

 - Table 5 states that the values given are means, but does not say how many samples each mean was generated from until Table 6. The contents of Table 6 should be in the figure captions; it is important to understand how many samples graphs are generated with
 - Similarly, the way that the shaded regions are calculated should be included up front in the first figure with them in it. At first I believed that the intervals were confidence intervals, but they are actually standard deviations.
 - How are the graph lines calculated? I am not sure, but they look like they have been smoothed out - the captions should indicate this if so. If they are smoothed, are the standard deviations calculated before or after smoothing?

Minor:
-------------
 - Figure 7 has only 3 random seeds compared. To make comparisons between the RND RNN and CNN policies methods you should use more seeds/samples.
 - On page 2 it is said that previous exploration methods are difficult to scale; a (very short) explanation on why would be appreciated
 - On page 4, it would be good to explain why one would be concerned that episodic rewards can leak information about the task to the agent
 - It would be interesting to plot the RND exploration bonus over time as training iteration progresses; this could give some insight into training dynamics that we cannot see from looking at reward trajectories alone.
 - It would be good to include experimentation around understanding if there is a benefit to using this technique in dense reward tasks.

---

> ### Author Response · Authors · 2018-11-21
> **Reply to reviewer 1 part 1**
>
> Thank you for your thoughtful feedback.
>
> We would like to begin with addressing the comparison of our method to existing exploration baselines. This has been the main concern expressed in your review, as well as a concern raised in a public comment.
>
> The previous SOTA result on Montezuma’s Revenge comes from Bellamare et al (2016). The technique is using a simple density model (CTS) to derive a pseudo-count bonus. They report two results, one with DQN as the policy optimizer and one with A3C. We ran an experiment with RND and 16 parallel actors (to match the 16 actors used in the A3C result). Below we compare the performance of RND (averaged over 5 seeds) with the published CTS results:
> RND at 150M frames: 4192
> DQN+CTS at 150M frames: 3705
> RND at 200M frames: 3831
> A3C+CTS at 200M frames: 1127
>
> As the comparison shows, RND’s performance is comparable to DQN+CTS at 150M frames of experience (it’s hard to know whether the difference is statistically significant due to large variance of results and larger instability of training with 16 actors compared to the setup used in our paper). We believe that the comparison to A3C+CTS is more meaningful, because both PPO and A3C are actor-critic methods collecting experience in the same way (from 16 parallel actors). In this comparison it is clear that PPO+RND performs better than A3C+CTS. We noticed that increasing the number of parallel actors trades off stability of training with sample efficiency, but even the results reported in the paper with 32 parallel workers are comparable or better than all previously reported results (mean score of 3263 at 150M frames of experience, 3688 at 200M).
>
> The purpose of our paper was to improve on scalable exploration methods. Among the scalable exploration bonuses, prediction-based bonuses are a prominent example. However these bonuses are subject to the noisy-TV problem, and so the focus of this paper was to address this problem. As such the dynamics-based baseline is the relevant baseline demonstrating the existence of the problem that we purport to fix.
>
> Below we provide more details on why density-based pseudocounts baselines (Bellemare et al, Ostrovski et al) are not as scalable as RND, which explains why we didn’t include them as baselines in our paper.
> _____
> Scalability of density-based pseudocounts, info-gain approximations, and RND.
>
> The second paragraph of introduction in our paper argues for the importance of scalability of modern deep RL methods.
>
> Pseudocount-based rewards are derived from the value of a density estimator of observed states before and after updating this density on the most recent observation. A difficulty of this approach is the computation of the pseudocounts on a batch of experience coming from parallel actors. There are several approaches to this computation. In one approach each actor maintains its own density model. This makes the computation embarrassingly parallel, but the memory requirement scales linearly and the different workers optimize different reward functions, which might diverge from each other. Another approach is to share a density model and update it sequentially in an arbitrary order. This makes memory requirements tractable, but the computation time scales linearly with the number of workers. Finally a compromise between these approaches shares the density model between workers, calculates the reward from a pre-update snapshot of the model for each experience in a batch in parallel, and then updates the model on the whole batch of experience. The memory requirements for this approach scale linearly with the number of workers. For this reason it would be impractical to run these baselines for billions of frames, especially for expressive density models with sizeable numbers of parameters.
>
> Approaches that approximate info-gain exploration bonus by comparing prediction errors before and after an update of a learned dynamics model are subject to the same fundamental scalability limitation.

---

> > ### Author Response · Authors · 2018-11-21
> > **Reply to reviewer 1 part 2**
> >
> > Next we would like to address your concerns on a lack of experimental details.
> > “The way that the value function is trained (i.e. the objective function) is never explained in the paper. The value function in PPO is typically (according to the baselines repository) trained at each step to fit (GAE advantage + previous value), but in the paper this is not elaborated on.”
> >
> > In this paper we omit the description of how PPO works because it is a standard method used in RL and we provide a reference to the original paper, as well as our implementation in the code accompanying the paper. As described in the original PPO paper we use generalized advantage estimation for fitting the baseline.
> >
> > “the statement that the extrinsic value function fits a stationary distribution on page 5 should be fixed”
> > Our statement is that “the extrinsic reward function is stationary”, which we believe is accurate. The targets for the value function estimate through GAE are indeed non-stationary (since the value function feeds into these estimates), but our statement is specifically about the stationarity of the reward function.
> >
> > “In Table 4 the $\lambda$ hyperparameter is listed, but is not described at all in the paper.”
> > We’ve added a reference to the PPO paper to table 4 to clarify the meaning of $\lambda$
> >
> > “though it is possible to infer, the paper never explicitly defines the intrinsic reward $i_t$ in the main paper text.”
> >
> > We’ve added the definition to the beginning of section 2.2.
> >
> > “The exact mechanism through which the "forward dynamics" baseline is never given” -
> > We believe that the third paragraph in section 3.6 describes the baseline in enough detail to reproduce it. The only difference from RND is that the target of the prediction problem is the features of the next state, rather than the current state, and that the action is additionally fed to the predictor network.
> >
> > More generally while we sympathize with your desire for completeness, the reason for somewhat concise descriptions of some of the technical points is the page limit on the main part of the paper. We had to balance the level of detail between addressing the intuitions behind our method, the interpretation of the experimental results, and descriptions of technical details. To aid the reader in understanding the technical details, we provide the full source code with the paper and moved other details to the appendix.
> >
> > “- Table 5 states that the values given are means, but does not say how many samples each mean was generated from until Table 6. The contents of Table 6 should be in the figure captions; it is important to understand how many samples graphs are generated with”
> >
> > We moved this information from captions to to the appendix in order to save space in the main text, but at your request we have moved this information back into the captions.
> >
> > “the way that the shaded regions are calculated should be included up front in the first figure with them in it”
> > We have moved this information into the beginning of section 3.
> >
> > “How are the graph lines calculated? I am not sure, but they look like they have been smoothed out - the captions should indicate this if so. If they are smoothed, are the standard deviations calculated before or after smoothing?”
> >
> > We have added a description of smoothing procedures to section 3.

---

> > > ### Comment · AnonReviewer1 · 2018-11-24
> > > **Followup author response**
> > >
> > > Thank you for your detailed responses. I will address responses by topic:
> > >
> > > Baselines
> > >
> > > Thank you for performing the experiments attempting to mimic the same conditions as previous work. Although I appreciate that the engineering effort is difficult, it would have been significantly better to integrate CTS with PPO directly to obtain a better baseline. I agree with the anonymous commenter that the direct comparison with A3C+CTS is not really fair; particularly considering that:
> > > - CTS+PPO seems to be specifically tuned for Montezuma’s Revenge vs Bellemare et al’s agent being tuned for a suite of games
> > > - A3C is generally a stronger method than PPO
> > >
> > > Overall, I strongly recommend that the authors include a comparison to the method of Bellemare et al with PPO in the paper; it is impossible otherwise for a reader to compare the effectiveness of the two methods.
> > >
> > > Finally, when comparing to previous work you should be extremely clear about the fact that previous methods used an order of magnitude less frames in attempting to solve the task. Including a discussion of the effect of many more frames is important - particularly with the experiments that you included above in your comment.
> > >
> > > Experimental Details
> > >
> > > Thank you for making many of the requested changes.
> > >
> > > The PPO paper only states that the algorithms presented use GAE, it never gives exactly how the value function is trained (just by reading the paper one would think that PPO uses the GAE paper’s value function training method, which is not at all similar to what PPO uses).
> > >
> > > As for the forward dynamics mechanism, reading the third paragraph in 3.6 does not give me a clear outline of the baseline mechanism; if you do not have room in the main text, it would be helpful to put a description of how the baseline is implemented (at a high level) in the appendix.
> > >
> > > More generally, I believe that this paper should be written in a way that it is relatively easy to follow without having to look into the accompanying code repository.

---

> > ### Public Comment · (anonymous) · 2018-11-21
> > **Comparing to Bellemare et al.'s A3C agent isn't really fair**
> >
> > Thanks for your replies. I'll comment here seeing as this is where you addressed my main concerns.
> >
> > I appreciate that there are some difficulties in applying CTS to PPO, although given that Bellemare et al. were able to apply it to A3C, which also involves parallel actors, I would have thought you could try the same approach. (They spell out exactly how they did it in Appendix D4.) As it stands, comparing against A3C's score of 1127 at 200M frames is misleading for a few reasons:
> > - Bellemare et al. apply their A3C agent to a wide range of games. Critically, this includes many dense reward games. To achieve better generality across the full suite, they scale the intrinsic reward down by a factor of 5. (See Appendix D4. Their A3C agent used beta = 0.01, whereas the DQN agent used beta = 0.05.) This undoubtedly hurt performance in sparse reward games. It's very unfair to compare an agent that was tuned for Montezuma's Revenge against one that was tuned for the full suite of games.
> > - PPO is generally stronger than A3C
> > - Bellemare et al. used gamma_E = 0.99, not 0.999. As your results show, this change makes a huge difference.
> >
> > I'm still concerned that the discount of 0.999 is overfit to sparse reward games. As you've said in the rebuttal to another reviewer "We tried gamma 0.999 because prior works using learning from demonstrations on Montezuma's Revenge had suggested that this was effective for Montezuma's Revenge". This only convinces me that you're fitting to one type of game. If gamma = 0.999 breaks the algorithm on dense reward games then you should really say so. It should be pretty easy to apply a vanilla PPO implementation with gamma = 0.999 to a few dense reward games and find out.
> >
> >
> > EDIT:
> > The additional results you've provided above just brought another matter to my attention:
> >
> > On page 5 you've said "Most experiments are run for 30K rollouts of length 128 per environment with 128 parallel environments, for a total of 1.97 billion frames of experience". But then in Figure 5, it looks like your strongest results were actually obtained with 1024 parallel environments, which equates to 16B frames (or TWO orders of magnitude more experience than Bellemare et al used).
> >
> > Throughout the rest of the paper, the figures don't show how many parallel actors were used, but based on the strength of the Montezuma scores in Figures 4 and 7, I'm assuming you used 1024 parallel environments. This should be made more explicit in all the places where you mention your headline scores. The statement that "most experiments" used 128 parallel environments threw me, because it's not true for your most significant results. I think this is a *really* interesting paper, but it worries me that future reviewers of competing papers are only going to see your headline score of 14,415 and not the training frames. In a research environment where papers unfortunately tend to be judged above all on achieving SOTA results, it's going to make it hard for small groups to compete, unless you make it clear that the paper is really about leveraging massive parallelism, not maximising sample efficiency. Clearly, the latter is something that still needs to be addressed in these kind of games.

---

### Public Comment · (anonymous) · 2018-11-14
**Are RND bonuses actually more effective than previous schemes?**

The results on Montezuma's Revenge are definitely very cool. However, I'm concerned that this one centrepiece result may be overshadowing several problems with the work in its current form.

First of all, I wish there was a proper comparison done between RND bonuses and previous state-of-the-art novelty bonus schemes. Unless I've misinterpreted the start of Section 3, your agents were trained for 1.97 *billion* frames of experience. This is about ten times more experience than Ostrovski et al.'s agents were trained on, so it's hard to tell whether distillation bonuses are actually any more effective than the neural density bonuses or Bellemare et al.'s CTS scheme. Furthering my suspicion, instead of comparing against these novelty schemes, which are well-known and were previously state-of-the-art on Montezuma, you've compared against bonuses from training a forward dynamics model, with the justification: "Burda et al. (2018) show that training a forward dynamics model in a random feature space typically works as well as any other feature space when used to create an exploration bonus." In reality though, Burda et al.'s results with this method on Montezuma are very underwhelming. It also bothers me that you've labelled your graphs in a way that obfuscates the amount of experience trained from. Don't get me wrong -- it's impressive that you've managed to train an agent to finish the first level of Montezuma -- but I suspect that this is mostly attributable to two factors (1) Running for more training time than previous agents (2) Setting the extrinsic reward discount to 0.999 instead of 0.99. Figure 4(a) seems to support this conclusion.

Another major concern I have with the paper is how much it focuses on one game. Montezuma’s Revenge seems like a best case for RND, because the vast majority of pixels are static background and the few enemies that exist follow set paths. Therefore, most pixel-level novelty is driven by the protagonist’s movement. As such, it is not surprising that "naive" novelty heuristics, such as CTS and RND, do well in this game. However, such schemes may struggle in games like Freeway, where there are a lot of moving entities. (Martin et al.’s 2017 agent struggled in Freeway because it was “awed” by all the different cars driving past -- see "Count-Based Exploration in Feature Space for Reinforcement Learning".) Again, certain choices that you've made only heighten my suspicion: In Ostrovski et al. (2017) they actually classify *seven* games as being sparse reward and hard exploration: Gravitar, Montezuma’s Revenge, Pitfall, Private Eye, Solaris, Venture and Freeway. Why did you select all of these games except for Freeway?

Breaking down your results on the other games tested doesn't do much to allay my concerns:
- Pitfall: None of the agents learn anything, which is no better or worse than in previous work.
- Private Eye: Ostrovski et al.'s PixelCNN agent reaches a score of around 15,000 points after around 30 million frames, whereas your agent takes over a billion frames to learn anything and doesn't beat this score.
- Solaris: From Figure 7, it looks like RND is detrimental, if anything.
- Gravitar: The RNN agent with RND is only marginally better than the RNN agent with RND. Further, "state-of-the-art" performance is only a result of training time. Ostrovski et al.'s Reactor-PixelCNN agent appeared to be on a very similar score trajectory at 150M frames.
- Venture: Again, "state-of-the-art" performance is only a result of training time. Ostrovski et al.'s Reactor-PixelCNN agent reached 1400 points by only 150M frames, and appears to be on a very similar score trajectory to your agent.

In Ostrovski et al.'s work, they also test their agent on many non-sparse games. While it is not expected that exploration-focused agents will excel in dense reward games, it is important to validate that they do not significantly underperform. In your work, one setting that I believe may be particularly overfit is gamma_E = 0.999. In sparse reward games, using a very mild discount is ok, because the returns will never “blow up”. However, in dense reward games, using such a mild discount will cause the returns to grow very large and thus potentially cause instability. I’m very curious how your configuration would perform on Video Pinball, for example. In your blog post, you've only showed how the agent performs on dense reward games when the extrinsic rewards are turned *off*, which smells like deliberate cherry picking. (To be clear, I'm not saying that you *have* cherry-picked, but I think you should try to avoid this perception.)

---

> ### Public Comment · (anonymous) · 2018-11-19
> **One final question if you don't mind**
>
> Throughout the paper and in the associated blog post, you've used the "noisy TV problem" as a motivating example. On page 8, you've noted that the dynamics-based agent gets stuck exploring the transition between rooms, because it can't accurately predict which room it will be in on the next frame. I can understand why RND avoids this problem: If the prediction network has seen room A and room B many times then it knows roughly what the random network is going to output in these rooms. However, if the agent is faced with *true* white noise, then won't the prediction error generally stay large? (Yes, given enough training time, it is true that the agent will have previously seen a noisy screen that is arbitrarily similar to the current one. Therefore, given enough representational capacity, the prediction error should *theoretically* go to zero. However, I really doubt that this is the case in practice. In Figure 1, it appears that even deep in training, there are still some states in the first room that yield a large prediction error. And this is the case despite the fact that the agent has seen similar screens many times before. In any event, you don't just need to show that the prediction error goes to zero on white noise -- you need to show that it becomes smaller than the prediction error elsewhere in the state space. Otherwise, the agent will still be encouraged to stare at the TV.)
>
> In your video on the blog post, it looks like the TV isn't actually showing white noise, but rather a random image from a fixed set. Again, I can understand that the RND agent will eventually learn the random network's encoding of each image in the set, so it will eventually get bored with looking at the TV in this case. Unless I'm wrong above though, I think you should remove the term "white noise" and replace it with the example of a TV showing a random image from a fixed set.

---

> > ### Author Response · Authors · 2018-11-21
> > **Reply to comment**
> >
> > We would like to clarify that what we meant by a noisy-TV problem was attraction of dynamics prediction-based exploration methods to stochastic transitions.  You are correct that a source of infinitely many states like white noise could be attractive to RND  (although whether the transitions are deterministic or stochastic in this case doesn't matter). We will update the example in the paper to reflect this.

---

> ### Author Response · Authors · 2018-11-21
> **Reply to comment**
>
> Thank you for comment, we have submitted replies to the reviewers which address many of your points.
>
> Regarding your question as to why we didn't run experiments on "Freeway", we did this simply because RL approaches have saturated performance on this game even without directed exploration.

---

> > ### Public Comment · (anonymous) · 2018-11-26
> > **Freeway**
> >
> > Regarding Freeway, I feel like you've slightly dodged my point. In your other recent paper on exploration ("Large-Scale Study of Curiosity-Driven Learning"), you *have* benchmarked on Freeway in Table 2. Furthermore, the intrinsic rewards look slightly harmful, even with a coefficient of 0.01 for the intrinsic rewards and 1.0 for the extrinsic rewards. My question is, how does your best Montezuma agent using RND perform on Freeway? I'm curious if the intrinsic reward swamps the extrinsic reward and ends up being really detrimental, or just a little bit. Since there's more going on visually in this game, my guess is that the RND bonuses may stay relatively large.

---

### Meta-Review · Area_Chair1 · 2018-12-14

**Confidence:** 5
**Recommendation:** Accept (Poster)

**Metareview:**

Pros:
- novel, general idea for hard exploration domains
- multiple additional tricks
- ablations, control experiments
- well-written paper
- excellent results on Montezuma

Cons:
- low sample efficiency (2B+ frames)
- unresolved questions (non-episodic intrinsic rewards)
- could have done better apples-to-apples comparisons to baselines

The reviewers did not reach consensus on whether to accept or reject the paper. In particular, after multiple rounds of discussion, reviewer 1 remains adamant that the downsides of the paper outweigh its good points. However, given that the other three reviewers argue strongly and credibly for acceptance, I think the paper should be accepted.